# *Lentinula edodes,* a Novel Source of Polysaccharides with Antioxidant Power

**DOI:** 10.3390/antiox11091770

**Published:** 2022-09-08

**Authors:** Tatiana Muñoz-Castiblanco, Juan Camilo Mejía-Giraldo, Miguel Ángel Puertas-Mejía

**Affiliations:** 1Grupo de Investigación en Compuestos Funcionales, Facultad de Ciencias Exactas y Naturales, Universidad de Antioquia UdeA, Calle 70 No. 52-21, Medellín 050010, Colombia; 2Facultad de Ciencias Farmacéuticas y Alimentarias, Universidad de Antioquia UdeA, Calle 70 No. 52-21, Medellín 050010, Colombia

**Keywords:** crude polysaccharides, mushroom, *Lentinula edodes*, antioxidant activity

## Abstract

The fruiting bodies of edible mushrooms represent an important source of biologically active polysaccharides. In this study, *Lentinula edodes* crude polysaccharides (LECP) were extracted in hot water, and their antioxidant and antiradical activities were investigated. The antioxidant activity of LECP was investigated against reactive species such as 1,1’-diphenyl-2-picrylhydrazyl, 2,2’-azino-bis(3-ethylbenzothiazoline-6-sulfonic acid, hydroxyl and superoxide anion radicals, reducing power with EC_50_ values of 0.51, 0.52, 2.19, 3.59 and 1.73 mg/mL, respectively. Likewise, LECP inhibited the lipid peroxidation induced in methyl linoleate through the formation of conjugated diene hydroperoxide and malondialdehyde. The main sugar composition of LECP includes mannose, galactose, glucose, fucose and glucuronic acid. Characterization by Fourier transform infrared spectroscopy and nuclear magnetic resonance determined that LECP was made up of α and β glycosidic bonds with a backbone of α-D-Glc, →6)-β-D-Glcp-(1→, →6)-α-D-Galp-(1→ and β-D-Manp-(1→ residues. The results showed that LECP can scavenge all reactive species tested in a concentration-dependent manner and with a protective effect in the initial and final stages of lipid peroxidation. The natural antioxidant activity of the LECP that was investigated strengthens the high medicinal and nutritional value of this mushroom.

## 1. Introduction

Mushrooms are widely cultivated worldwide due to their high nutritional value and medicinal properties [1,2]. The cultivation of mushrooms improves their ecological value, promotes sustainable use and helps the conservation of the environment [3]. Edible mushrooms are of interest in the food industry because they have been recognized as functional foods due to their sensory characteristics (aroma and flavor) and their favorable chemical composition, content in carbohydrates, dietary fiber, proteins, vitamins and minerals. Mushroom consumption has been shown to help prevent different types of cancer and heart disease, among other benefits [4,5]. Mushrooms contain bioactive compounds such as polysaccharides, proteins and small organic molecules (secondary metabolites) such as polyphenols, as well as polysaccharide–protein and polysaccharide–phenol complexes, which have been recognized as the main bioactive components of medicinal mushrooms [6].

*Lentinula edodes* (Shiitake mushrooms) is one of the most cultivated edible mushrooms [7]. Recently, bioactive polysaccharides obtained from *L. edodes* have shown an important role as free radical scavengers in the prevention of oxidative damage and can be explored as for its new potential as a natural antioxidant. Due to public concern about the adverse effects of synthetic chemical antioxidants on human health, there has been a growing interest in the discovery and application of natural antioxidants. Although *L. edodes* crude polysaccharides (LECP) have been known for their biological potential in recent years, difficulties in their extraction, purification and chemical composition have been an obstacle to their consumption and marketing. Therefore, a basic understanding of lentinan bioactivities is essential for its successful application in disease prevention and treatment [8,9].

In this work, we have extracted and characterized polysaccharides from *L. edodes* fruiting bodies in hot water and evaluated their antioxidant activity. First, the chemical composition, monomer units and glycosidic bonds of the crude polysaccharides were investigated by spectroscopic and chromatographic analyses. The evaluation of the antiradical and antioxidant activity of the crude polysaccharides demonstrated the synergistic relationship between the different components of the hot water soluble extract and the type of glycoside bond to counteract radical and oxidant species. Thus, water-soluble polysaccharides are a source of natural antioxidants and can promote the consumption of the *L. edodes* mushroom as a functional food.

## 2. Materials and Methods

### 2.1. Materials

The fruiting bodies of the *L. edodes* mushroom were obtained from the Santa Rita farm, Cardonal Village, located in the town of Cogua, Cundinamarca, Colombia, which is certified in Good Agricultural Practices by the Colombian Agricultural Institute (ICA) in accordance with ICA resolution 30021 of 2017 and 82394 of 2020 and Otro sí No. 13, Framework Contract for Access to Genetic Resources and their Derived Products, 126 of 2016, RGE0156-13, Ministerio de Ambiente y Desarrollo Sostenible, Colombia. 1,1-diphenyl-2-picrylhydrazyl (DPPH), 2,2’-azino-bis(3-ethylbenzothiazoline-6-sulfonic acid) (ABTS), pyrogallol, trifluoroacetic acid (TFA), 1-phenyl-3-methyl-5-pyrazolone (PMP), glucose (Glc), galactose (Gal), mannose (Man), fucose (Fuc) and glucuronic acid (GlcA), chondroitin sulfate, carbazole, 1,9-dimethyl-methylene blue, bovine serum albumin (BSA), Bradford reagent, gallic acid, potassium persulfate (K_2_S_2_O_8_), congo red, tris(hydroxymethyl)aminomethane, methyl linoleate (MeLo), butylated hydroxytoluene (BHT), 2-thiobarbituric acid (TBA), Iron (III) chloride (FeCl_3_), potassium ferricyanide (III) (K_3_[Fe(CN)_6_]), sodium tetraborate and deuterium oxide (D_2_O) were obtained from Sigma-Aldrich (St. Louis, MO, USA). Ascorbic acid (AA) and hydrogen peroxide H_2_O_2_ were acquired from PanReac AppliChem (Darmstadt, Germany). Ethanol, acetone, ether, chloroform, butanol, methanol, acetonitrile, hydrochloric acid (HCl), phenol, Folin–Ciocalteu reagent, salicylic acid, ferrous sulfate (FeSO_4_), sulfuric acid (H_2_SO_4_) and sodium hydroxide (NaOH) were purchased from Merck Millipore (Darmstadt, Germany). All reagents were used without any further treatment.

### 2.2. Extraction of Polysaccharides from the Fruiting Bodies of L. edodes

The extraction of water-soluble polysaccharides was carried out according to previously described methodologies with some modifications [10,11]. Briefly, the dried mushroom was extracted with ethanol for 3 h to remove lipids, pigments, polyphenols and other small molecules. The residue was extracted twice with distilled water (residue of ethanolic extraction: distilled water = 1:20 *w*/*v*) at 95 °C for 3 h. The supernatants were concentrated under vacuum rotary evaporation at 50 °C and then mixed with Sevag reagent (n-butanol: chloroform = 1:4, *v*/*v*) for 30 min (Sevag reagent: polysaccharide solution = 5:1, *v*/*v*). Then, they were centrifuged at 3500 rpm for 15 min, and the process was repeated 2 times. The supernatant was precipitated with ethanol at 4 °C overnight (ethanol: polysaccharide solution ratio 5:1, *v*/*v*). The precipitated polysaccharides were collected by centrifugation at 3500 rpm for 15 min and were then washed twice with ethanol, acetone and ether, respectively. The polysaccharides were dissolved in distilled water and dialyzed against distilled water in a dialysis membrane (MWCO12-14 KDa) for 48 h. Finally, the resulting solution was freeze-dried, and the *L. edodes* crude polysaccharides (LECP) were obtained. The estimation of the molecular weight of the polysaccharides of the crude extract obtained was performed by ultracentrifugation using 10–50, 50–100 and >100 KDa MWCO filters for 30 min at 5000 rpm and finally by lyophilizing the fractions obtained.

### 2.3. Characterization of LECP

#### 2.3.1. Sugar and Glucans Composition

The neutral sugar content was measured by the phenol–sulfuric acid method [12]. An amount of 0.03 mL of LECP (1 mg/mL) was mixed with 0.370 mL of distilled water, and 2 mL of H_2_SO_4_ was added. Then, the solution was shaken, and 0.40 mL of phenol (5% *w*/*v*) was added. The solution was shaken, heated at 90 °C for 5 min and cooled in a water bath. The absorbance was measured at 490 nm. The uronic sugar content was determined by the carbazole–sulfuric acid method [13]. An amount of 0.08 mL of LECP (1 mg/mL) was mixed with 0.32 mL of distilled water, and 2 mL of sodium tetraborate in H_2_SO_4_ (0.095% *w*/*v*) was added. The solution was stirred and heated at 100 °C for 12 min. Then, 0.04 mL of the carbazole ethanolic solution (0.2% *w*/*v*) was added, and the solution was stirred and heated at 100 °C for 10 min. The solution was cooled in a water bath, and the absorbance was measured at 525 nm. The sulfated content of sugar was analyzed by the dimethylmethylene blue method [14]. An amount of 0.10 mL of LECP (1 mg/mL) was mixed with 0.40 mL of distilled water, and 3 mL of 1,9-dimethyl-methylene blue (0.0011% *w*/*v*, dissolved in sodium acetate 0.05 M, pH 4.75), was added. The solution was shaken and incubated in the darkness for 30 min. The absorbance was measured at 525 nm. The β-glucan content of LECP was measured using a mushroom and yeast-specific β-glucan kit (Megazyme International, Wicklow, Ireland) following the manufacturer’s instructions.

#### 2.3.2. Protein and Phenol Composition

The total protein content was determined by the Bradford method [15]. An amount of 0.25 mL of LECP (1 mg/mL) was mixed with 0.75 mL of Bradford reagent and stood in the darkness for 5 min. The absorbance was measured at 595 nm. The total content of the phenols was determined by the Folin–Ciocalteu method [6]. An amount of 0.10 mL of LECP (1 mg/mL) was mixed with 1.50 mL of distilled water and 0.10 mL of Folin–Ciocalteu reagent and stood for 10 min at room temperature. Next, 0.30 mL of sodium carbonate (20% *w*/*v*) was added, and the mixture was incubated at 40 °C for 30 min. The absorbance was measured at 765 nm. Information about the reference substance and the regression equations for each method used are shown in Table 1.

#### 2.3.3. Ultraviolet and FT-IR Spectroscopy Analysis

The ultraviolet absorption spectra of the LECP aqueous solution were measured by a Thermo Scientific Evolution 60S UV-Visible spectrophotometer between 200–500 nm. The FT-IR spectra were obtained in the PerkinElmer Spectrum Two Spectrometer by the total attenuated reflection (ATR) technique between 500 and 4000 cm^−1^.

#### 2.3.4. Monosaccharide Composition

The monosaccharide composition of LECP was determined by a PMP high-performance liquid chromatography-diode-array detector (HPLC-DAD) [6]. Briefly, 5 mg of LECP was hydrolyzed in 2 mL of TFA (2.0 M) at 110 °C for 4 h. The hydrolysate was dried by vacuum evaporation at 50 °C, and the residue was dissolved in methanol to remove residual TFA. The methanol was evaporated, and this step was repeated four times [16]. Then, the residue was dissolved in 2 mL of water. An amount of 0.45 mL of hydrolysate solution, 0.45 mL of PMP (0.5 M) methanolic solution and 0.45 mL of NaOH (0.3 M) were mixed and incubated at 70 °C for 30 min. Next, 0.45 mL of HCl was added to stop the reaction. The product was partitioned with chloroform three times and was filtered through a 0.45 µm membrane. HPLC analysis was carried out using an LC 300 HPLC System equipped with DAD detectors and a Pinnacle C18 (250 mm × 4.6 mm). The mobile phase consisted of potassium phosphate buffer saline (0.05 M, pH 6.9) with 83% (solvent A) and 17% acetonitrile (solvent B), and the wavelength for UV detection was 250 nm. Glucose, galactose, mannose, fucose and glucuronic acid were used as reference substances.

#### 2.3.5. NMR Analysis

LECP (30 mg) was dissolved in D_2_O (0.5 mL). The 1D (^1^H, and DEPT 135) and 2D (^1^H–^13^C HSQC) spectra were measured using a Bruker Ascend III HD 600 MHz spectrometer at 25 °C.

#### 2.3.6. Preliminary Triple Helix of the Conformation by Congo Red Analysis

The triple helix structure of LECP was confirmed by the Congo red test with some modifications [17]. An amount of 0.5 mL of LECP (1 mg/mL) was mixed with 0.5 mL of Congo red solution (80 mM), and then a NaOH (1.0 M) solution was gradually added to the mixtures to a final NaOH concentration of 0, 0.1, 0.2, 0.3, 0.4 and 0.5 M. The solution without polysaccharide was used as a negative control. After reacting for 5 min, the maximum absorption wavelength in the range of 400 to 600 nm was measured.

### 2.4. Antiradical Activity

#### 2.4.1. DPPH Radical Scavenging Assay

The DPPH radical scavenging activity was measured by the method previously reported with some modifications [9]. An amount of 0.2 mL of LECP with different concentrations (0.05–4.0 mg/mL) and 1.0 mL of DPPH (0.05 mM) methanolic solution were shaken and incubated in the darkness for 30 min. Then, the absorbance of the mixtures was measured at 517 nm using a Thermo Scientific Evolution 60S UV-Visible spectrophotometer. Ascorbic acid (AA) was used as the positive control. The DPPH radical scavenging activity was calculated as:(1)$$\mathrm{DPPH}\;\mathrm{radical}\;\mathrm{scavenging}\;\mathrm{activity}\;\left(\%\right)=\left(\;\frac{A_{\mathrm{DPPH}\;}-{\;A}_{S\;}}{A_{\mathrm{DPPH}\;}}\right)\;\times \;100$$
where A_DPPH_ is the is the absorbance of the DPPH methanolic solution, and A_S_ is the absorbance of the sample (DPPH methanolic solution and LECP at a particular concentration).

#### 2.4.2. ABTS Radical Scavenging Assay

The ABTS radical scavenging activity of LECP was determined according to a previous report with some modifications [18]. Briefly, 2.5 mL of ABTS (7 mM) and 2.5 mL of K_2_S_2_O_8_ (2.45 mM) were mixed and kept in the dark at room temperature for 12–16 h before use. The solution was diluted with distilled water to a final absorbance of 0.70 ± 0.02 at 734 nm. Afterwards, 1.0 mL of diluted ABTS solution and 0.1 mL of LECP with different concentrations (0.2–3.0 mg/mL) were mixed and incubated in the darkness for 6 min. The absorbance was measured at 734 nm, and AA was used as the positive control. The ABTS scavenging activity was given by:(2)$$\mathrm{ABTS}\;\mathrm{radical}\;\mathrm{scavenging}\;\mathrm{activity}\;\left(\%\right)=\left(\;\frac{A_{\mathrm{ABTS}\;}-{\;A}_{S\;}}{A_{\mathrm{ABTS}\;}}\right)\;\times \;100$$
where A_ABTS_ is the absorbance of the diluted ABTS solution, and A_S_ is the absorbance of the sample (diluted ABTS solution and LECP at a particular concentration).

#### 2.4.3. Hydroxyl Radical Scavenging Assay

The hydroxyl radical scavenging assay was carried out according to a previous report [18]. Concisely, 0.1 mL of LECP with a different concentration was mixed with 0.5 mL of FeSO_4_ (0.15 mM), 0.5 mL of H_2_O_2_ (6 mM), 0.2 mL of salicylic acid (2 mM, dissolved in ethanol) and 0.2 mL of distilled water. After that, the mixtures were incubated at 37 °C for 1 h and were measured at 510 nm. AA was used as the positive control. The hydroxyl radical scavenging activity was calculated as:(3)$$\mathrm{Hydroxyl}\;\mathrm{radical}\;\mathrm{scavenging}\;\mathrm{activity}\;\left(\%\right)=\left(1-\frac{\left(A_{i}-A_{\mathrm{ii}}\right)}{A_{0}}\right)$$
where A_0_ is the control absorbance without the sample; A_i_ is the absorbance of the sample; and A_ii_ is the background absorbance without H_2_O_2_.

#### 2.4.4. Superoxide Radical Scavenging Activity

The superoxide radical scavenging assay was executed according to a previous report [19]. Briefly, 0.125 mL of LECP with different concentrations (0.5–10.0 mg/mL) and 1.125 mL of Tris–HCl buffer (50 mM, pH = 8.2) were incubated at 25 °C for 20 min. After that, 0.250 mL of pyrogallol (25 mM) was added to the mixture and incubated at the same temperature for 4 min. The reaction was stopped by adding 0.250 mL of HCl (8 mM). AA was used as the positive control. The absorbances of the mixtures were measured at 325 nm, and the superoxide radical scavenging activity was calculated as:(4)$$\mathrm{Superoxide}\;\mathrm{anion}\;\mathrm{radical}\;\mathrm{scavenging}\;\mathrm{activity}\;\left(\%\right)=\left(1-\frac{\left(A_{i}-A_{\mathrm{ii}}\right)}{A_{0}}\right)$$
where A_0_ is the control absorbance without the sample; A_i_ is the absorbance of the sample; and A_ii_ is the background absorbance without the pyrogallol solution.

#### 2.4.5. Reducing Power Assay

The reducing power was assessed according to a previously reported method [20]. An amount of 0.250 mL of LECP with different concentrations (0.2–4.0 mg/mL) was mixed with 0.625 mL of phosphate buffer (200 mM, pH 6.6) and 0.625 mL of K_3_[Fe(CN)_6_] (1%, *w*/*v*) solution. The mixtures were incubated for 30 min at 50 °C. Then, 0.625 mL of TFA (10%, *w*/*v*) was added, and the reaction mixtures were centrifuged for 10 min at 5000 rpm. Next, 0.625 mL of supernatant was mixed with 0.625 mL of distilled water and 0.125 mL of FeCl_3_ (1%, *w*/*v*). The mixtures were kept for 10 min in the dark, and the absorbance of the reaction mixture was measured at 700 nm. AA was used as the positive control. The reducing power was calculated as:(5)$$\mathrm{Reducing}\;\mathrm{power}\;\left(\%\right)=A_{i}-A_{\mathrm{ii}}$$
where A_i_ is the absorbance of the sample, and A_ii_ is the absorbance of the sample without FeCl_3_.

### 2.5. Antioxidant Activity

The antioxidant activity was investigated by the inhibition of lipid peroxidation in MeLo according to a previous report [21]. An amount of 0.9 mL of MeLo (10 mM) and 0.1 mL of LECP or BHT (2 mg/mL) were exposed to accelerated oxidation for 5 days at 40 °C in test tubes. Later than thermal oxidation, each sample was dissolved in 1 mL of ethanol.

#### 2.5.1. Conjugated Diene Hydroperoxide (CDH)

The concentration of CDH produced during oxidation was measured at 234 nm. The mixtures were diluted with ethanol in a ratio of 1:25. An extinction coefficient of 29,000 M^−1^ cm^−1^ was used. The peroxidation level was expressed as mmol CDH kg^−1^ MeLo.

#### 2.5.2. Thiobarbituric Acid Reactive Substances (TBARS)

The level of lipid peroxidation was also expressed as the malondialdehyde (MDA) content and was determined as TBARS. Briefly, 0.05 mL of the sample, 0.350 mL of ethanol, 0.100 mL of BHT-ethanol and 0.500 mL of TBA (0.37%) in HCl (0.25 mM) were incubated at 90 °C for 30 min. Then, the mixtures were cooled in an ice bath and centrifugated at 3000 rpm for 10 min. The absorbances of the samples were measured at 535 nm and were corrected for non-specific turbidity by subtracting the absorbance at 600 nm. MeLo was used as the control, and the peroxidation was expressed as mmol MDA kg^−1^ MeLo using a molar extinction coefficient of 156.000 M^−1^ cm^−1^.

### 2.6. Statistical Analysis

The experimental results included three replications, and the data were expressed as mean ± standard deviation (SD). The data were analyzed by an ANOVA (*p* < 0.05). The extract concentration providing 50% of the radical scavenging activity (EC_50_) was calculated from the graph of radical scavenging activity percentage against LECP concentration.

## 3. Results and Discussion

### 3.1. Extraction and Chemical Characterization of LECP

#### 3.1.1. Chemical Composition and Monosaccharide Composition

LECPs were extracted with hot water from the fruiting bodies of *L. edodes* and were purified by dialysis. The yield of the LECPs was 2.3% of the mushroom dry weight. The LECPs were mainly composed of 70.23 ± 2.03% neutral sugars, 8.81 ± 1.49% acid sugars, 3.41 ± 0.11% proteins and a low content (2.83 ± 0.08%) of phenolic compounds. The LECPs were composed of 16.94 ± 0.66% total glucans, 13.86 ± 0.79% β-glucans and 3.07 ± 0.14% α-glucans. The content of beta and alpha glucans was comparable with previous reports of extracts from *L. edodes*, which have a marked impact on their biological activity [22,23]. The beta glucans found in LECP do not represent more than 20% of the extract, suggesting that not only are the beta-glucans present in the extracts responsible for the antioxidant activity, but there are also other monomers that are involved in the bioactivity. This is related to reports of heteropolysaccharides with different types of unions that present high biological activities [24,25].

Although the LECPs were precipitated with ethanol and purified with membrane dialysis (MWCO12-14 KDa) to remove low molecular weight components from the water extract, a small (<3.0%) fraction of phenolic compounds was still present in the LECPs, due to the interactions of phenolic compounds with polysaccharides with high molecular weights. This is related to the possibility to form hydrogen bonding or hydrophobic interactions with polysaccharides [6]. Likewise, the proteins present in the LECP may have interacted with the polysaccharides through covalent bonds, forming polysaccharide–protein (PSP) complexes such as proteoglycans [26]. The LECPs did not present sulfated sugars, as previously reported [27]. The chemical composition of the polysaccharides varied in comparison with previously reported extracts, since the content of bioactive compounds of the mushrooms depends on factors such as age, stage of development, cultivation techniques and the substrate of the strain, among other factors [28]. The range of the molecular weights of LECPs was 80% polysaccharides >100 kDa, 10% polysaccharides from 50 to 100 kDa and 10% polysaccharides from 10 to 50 kDa. These results indicated that LECP is composed of polysaccharides of high molecular weights. This could be related to its antioxidant activity, since it has been reported that polysaccharides with high molecular weights could have more side chains, and this can increase the exposure to functional groups such as C-H and O-H that contribute to antioxidant ability [29].

HPLC-DAD with pre-column PMP derivatization combined with hydrolysis by TFA was used for the determination of the monomeric profiles of LECP. The derivatization protocol was suitable for the simultaneous analysis of neutral and acidic sugars and was performed under relatively mild conditions without the use of specialized columns [30]. According to the chromatogram of mixed standard monosaccharides shown in Figure 1, the LECP was composed mostly of Man, Glc, Gal and Fuc with a low content of GlcA. This result demonstrates the high content of neutral sugars and the small fraction of uronic sugars of LECP, as indicated in the chemical composition analysis.

#### 3.1.2. UV-Vis Spectra and FT-IR Spectra Analysis of LECP

The UV-Vis spectrum of the LECPs is shown in Figure 1A. The spectrum shows typical absorption bands between 260 and 280 nm associated with proteins and phenols present in the sample, according to the results of the composition analysis. Absorbance within the range of 260–280 nm is commonly attributed to π − π* electron transitions in aromatic and polyaromatic compounds found in most conjugated molecules, including proteins [31,32]. Although the Sevag method was used to deproteinize the LECPs, the protein content detected was 3.41%, suggesting that it could be a protein-bound polysaccharide. It has been reported that the antioxidant activity of protein-bound polysaccharides is enhanced by a synergistic effect [33].

The FT-IR spectrum (Figure 1B) of LECP showed typical absorption peaks of polysaccharides. The absorption of the broad peak at 3240 cm^−1^ was related to the O–H stretching vibration of the strong inter and intramolecular interactions of the polysaccharide chain. The band around 2890 cm^−1^ corresponded to the C-H stretching vibration. The absorption bands at 1640 and 1518 cm^−1^ were related to the presence of proteins and aromatic compounds, such as aromatic polyphenols (C=C and C=O stretching vibrations) [34]. The absorption band around 1390 cm^−1^ corresponded to OH groups of phenolic compounds [34,35]. These results were consistent with the composition results of the LECPs. The band at 1643 cm^−1^ was attributed to the stretching vibration of the C=O groups [36]. The peak at 1030 cm^−1^ indicated the presence of pyranose rings in the LECPs. Furthermore, absorption peaks at 912 and 864 cm^−1^ indicated the presence of β- and α-glycosidic linkages, respectively [9,37]. After that, it was deduced that the LECPs contained both α- and β- configurations.

#### 3.1.3. NMR Analysis

NMR spectra include ^1^H, DEPT-135 and HSQC spectra. Resonances were assigned according to values from the literature [38,39,40]. In the ^1^H NMR spectrum, four anomeric hydrogen signals were observed in the range of δ 4.4–5.4 ppm. This was consistent with the FTIR results, which showed that LECP contained α and β glycosidic bonds.

The ^1^H NMR spectrum showed the presence of α-d-glucopyranosyl, α-d-galactopyranosyl, β-d-mannopyranosyl and β-d-glucopyranosyl at 5.31, 5.06, 4.74 and 4.46 ppm, respectively (Figure 2A). These results were confirmed by the analysis of the monosaccharide composition for LECP. In the HSQC spectrum (Figure 2C), four cross-peaks (A–D) were observed at 5.31/99.29, 4.46/102.8, 5.06/98.12 and 4.74/101.63 ppm. They were assigned to H1/C1 of α-d-Glc-(1→, →6)-β-d-Glcp-(1→, →6)-α-d-Galp-(1→ and β-d-Manp-(1→, respectively (see Table 2). The DEPT-135 spectra shown in Figure 2B confirmed the C-6 linkages of residues in the HSQC spectrum.

#### 3.1.4. Preliminary Results on Triple-Helix Conformation

The biological and functional activities of polysaccharides are associated with their triple-helix conformation [41]. Polysaccharides with a triple-helix structure have been shown to react with Congo red to form complexes. The maximum wavelength absorption (*λ_max_*) of the complex is red shifted in comparison to Congo red [42]. The changes in the *λ_max_* for Congo red and Congo red with LECP at different concentrations of NaOH are shown in Figure 1D. The results showed that LECP formed complexes with Congo red in alkaline solution, and the value of *λ_max_* increased. When the NaOH concentration was 0.4 M, no significant decrease in the *λ_max_* value was detected, which could be related to the strong interchain hydrogen bonds [43]. This confirmed the presence of a triple helix conformation in LECP.

### 3.2. Antiradical Activity

The antiradical activity of LECP and AA was assessed for different in vitro assays. The EC_50_ value was defined as the concentration of antioxidant necessary to reduce 50% of the initial free radical [9,44]. A lower EC_50_ value means higher antiradical activity of the polysaccharide extract. The results of the DPPH, ABTS, hydroxyl and superoxide radical scavenging activity, as well as the reducing power expressed as EC_50_ (mg extract/mL) values, are shown in Table 3.

#### 3.2.1. DPPH Radical Scavenging Ability

DPPH is stable organic nitrogen radical with an unpaired electron, which can reduce by accepting an electron or hydrogen [18,45]. The DPPH radical scavenging ability of LECP is shown in Figure 3A. LECP showed a dose-dependent radical scavenging ability. In the higher concentration (1.6 mg/mL), LECP exhibited very high radical scavenging activity (89.63%), which was close to that of AA (97.16%). Nevertheless, in the lower concentrations (0.05 to 0.6 mg/mL), the radical scavenging ability of LECP was lower than that of AA. The EC_50_ value of LECP (0.51 ± 0.05 mg/mL) was lower compared to that reported for *Lentinus edodes* (2.16 mg/mL) [9], *Pleurotus ostreatus* (2.36 mg/mL) [46] and *Pleurotus djamor* (3.83 g/mL) [47] polysaccharides, indicating the great potential of LECP to scavenge DPPH radicals. This enhanced activity of LECP could be related to the presence of proteins, since it has been shown that there is a synergistic effect in the ability to eliminate the DPPH radical between polysaccharides and proteins [33,48].

#### 3.2.2. ABTS Radical Scavenging Ability

The antiradical activity of polysaccharides was measured by an ABTS assay. Polysaccharides can donate an electron or a hydrogen atom to an unstable ABTS radical to form a stable ABTS radical [18]. As shown in Figure 3B, the ABTS radical scavenging activity of LECP increased in a concentration-dependent manner. LECP scavenged 16.70% and 35.40% ABTS radicals, respectively, at concentrations of 0.1 and 0.3 mg/mL, and the activity increased to 57.12% at a concentration of 0.6 mg/mL. Although the ABTS radical scavenging ability increased with polysaccharide concentration, it was lower than that of AA. When the concentration reached 3.0 mg/mL, the scavenging rate of the ABTS radical was 95.30%, which was close to that of AA (99.72%). The EC_50_ value of LECP (0.52 ± 0.02 mg/mL) was lower compared to that reported for *Lentinus edodes* (2.17 mg/mL) [9] and *Pleurotus djamor* (0.82 g/mL) [47] polysaccharides. These differences in the ABTS radical scavenging activity could be related to the active hydroxyl groups associated with monosaccharides [20,49] and the total content of polyphenols [45].

#### 3.2.3. Hydroxyl Radical Scavenging Ability

Hydroxyl radicals are one of the most reactive free radicals, as they can pass through cell membranes and react with DNA, proteins, lipids and carbohydrates [18,47]. The hydroxyl radical scavenging ability of LECP is shown in Figure 3C. It could be observed that both LECP and AA showed antiradical ability in a concentration-dependent manner. The OH radical scavenging rate of LECP significantly increased from 23.41% to 54.81% as the LECP concentration increased from 0.2 to 2.5 mg/mL, and the OH radical scavenging rate of AA increased from 28.04% to 96.85% during the same concentration range. The EC_50_ value of LECP (2.19 ± 0.18 mg/mL) was approximately twelve times lower than that reported for *Pleurotus ostreatus* (27.86 mg/mL) [46], demonstrating the great potential of LECP to scavenge hydroxyl radicals compared to other edible mushrooms. The possible mechanism of the hydroxyl scavenging ability could be associated with the number of active hydroxyl groups in the molecule [45,50]. Therefore, the combination of the high amount of glucose and mannose of the polysaccharides linked by α and β glycosidic bonds, as shown in the HPLC and NMR analyses, together with the presence of phenolic compounds coupled to LECPs, gives more active hydroxyl groups to improve the ability to scavenge hydroxyl radicals.

#### 3.2.4. Superoxide Radical Scavenging Ability

Superoxide anion radicals (O_2_^•−^) have a longer lifetime than other radicals, although they are less reactive and can participate in other reactions, leading to the formation of other reactive oxygen species, which, in excess, can disrupt the balance of the organism and can promote DNA damage and the spread of various diseases [50,51].

In Figure 3D, it is shown that the superoxide anion scavenging activity of LECP and AA followed a concentration-dependent manner. The scavenging rate increased from 17.22% to 59.91% as the LECP concentration increased from 0.5 mg/mL to 5.0 mg/mL. Compared with LECP, AA showed a higher scavenging ability (98.02%) at 5.0 mg/mL. The EC_50_ value of LECP (3.59 ± 0.06 mg/mL) was similar to that reported for *Lentinus edodes* polysaccharides (3.45 mg/mL) [9]. The potential of polysaccharides to scavenge superoxide radicals could be related to the presence of electrophilic groups in their molecular structure that facilitate the release of hydrogen from the OH bond to stabilize the superoxide radical. As the number of electron-withdrawing groups attached to the polysaccharide becomes greater, the energy required to dissociate the OH bond becomes weaker. Thus, an increase in the concentration of LECP and the presence of some electrophilic groups such as keto or aldehyde groups could increase the scavenging activity of the superoxide radical [50,51]. The presence of uronic acids in LECP provides O_2_^•−^ scavenging activity [52,53], highlighting the health benefit of the crude extract of *Lentinula edodes* as a natural source of antioxidants.

#### 3.2.5. Reducing Power Assay

The reducing power can be considered an indicator of antioxidant capacity, since it has been described that there is a direct relationship between the antioxidant activity and the reducing power of polysaccharides [48]. The reducing power can be evaluated by the potassium ferricyanide reduction method by electron donation in the reduction of [Fe^3+^(CN)_6_]^3−^ into [Fe^2+^(CN)_6_]^4−^ [20,45]. As shown in Figure 3E, at a concentration of 0.4 mg/mL, the absorbance value of LECP was 0.159, but in the case of AA, the value was 2.18. The LECP showed an absorbance value of 0.56 at 2.0 mg/mL, whereas the AA showed a value of 2.5 at the same concentration. The reducing power of the LECP increased with the increase in concentration. At higher concentrations, the absorbance value of LECP oscillated at approximately between 2.5 and 2.6, whereas the value of LECP increased. These results suggest that the reducing power of LECP is directly associated with the ability to donate electrons from reducing sugars and polyphenols coupled to the polysaccharide chain, which are capable of reducing the [Fe^3+^(CN)_6_]^3−^ ion to generate the colored complex.

### 3.3. Antioxidant Activity

The antioxidant activity was assessed by the inhibition of lipid peroxidation in MeLo through the production of CDH and TBARS. The results obtained by the lipid model (See Table 4) showed that, in the MeLo assay with BHT and LECP, there was a protective effect in the early stages of lipid peroxidation, which was evidenced by the low formation of CDH (14.36 ± 2.41 mmol/Kg MeLo). A protective effect was also observed in the final stages of lipid peroxidation, which was confirmed with the decrease in the formation of MDA (0.08 ± 0.02 mmol/Kg MeLo) compared to MeLo (10.42 ± 0.95 mmol/Kg MeLo). LECP presented an inhibition in the formation of CDH and MDA similar to that observed with BHT. These results demonstrated the remarkable inhibition of lipid peroxidation by scavenging lipid-derived radicals from LECPs. This activity can be related to the acidic sugars, proteins and polyphenols present in LECP, which indicates the food and pharmaceutical relevance of mushroom extracts. The LECP lipid peroxide radical scavenging assay showed agreement with extracts of edible mushrooms of *Oudemansiella radicata*, which needs 1 mg/mL to exceed 50% inhibition of lipid peroxidation [54]. Thus, extracts of Lentinula edodes are the most promising lipid peroxidation inhibitors reported previously [55,56].

Although the mechanism by which antioxidant activity occurs is still not understood, it has been reported that the biological activity of polysaccharides is influenced by factors such as the composition of monosaccharides, molecular weight and degree of branching, among other factors. The higher antioxidant capacity of crude polysaccharides could be due to the presence of compounds such as uronic sugars, phenols and proteins [48,57]. Therefore, not performing purification processes of the crude extracts can improve their antioxidant activity and make the process of obtaining them more favorable in terms of time and cost. Furthermore, the results obtained show that LECP could have a potential use as a natural antioxidant and could be used as a functional component in the food industry. The results of this work can allow the promotion of the cultivation and consumption of mushrooms as an important source of compounds with potential use as antioxidants.

## 4. Conclusions

Crude water-soluble polysaccharides from the cultivated mushroom *Lentinula edodes* were successfully extracted. LECP mainly contained polysaccharides of the type (1→6)-β-d-glucan, (1→3)-β-d-glucan, (1→3)-α-d-glucan and (1→3),(1→6)-β-d-glucan, proteins, polysaccharide–protein conjugates and a small amount of phenolic compounds according to α and β-glucan, FTIR, HPLC and NMR analyses. The Congo red colorimetric assay confirmed the triple-helix conformation of LECP. The inhibition of lipid peroxidation induced in methyl linoleate, by the formation of conjugated diene hydroperoxide and malondialdehyde, confirmed the excellent antioxidant capacity of LECP. The antioxidant and antiradical activity of LECP was closely related to neutral sugars linked by α and β glycosidic bonds, acid sugars, polysaccharide–protein conjugates and polysaccharide–polyphenols, which allow for the efficient elimination of radical species with a high reducing power. In this way, LECP is a source of natural antioxidants with health benefits as a functional food with potential in the food and pharmaceutical industries, highlighting the cultivation and consumption of mushrooms in Colombia and the world.

## Figures and Tables

**Figure 1 antioxidants-11-01770-f001:**
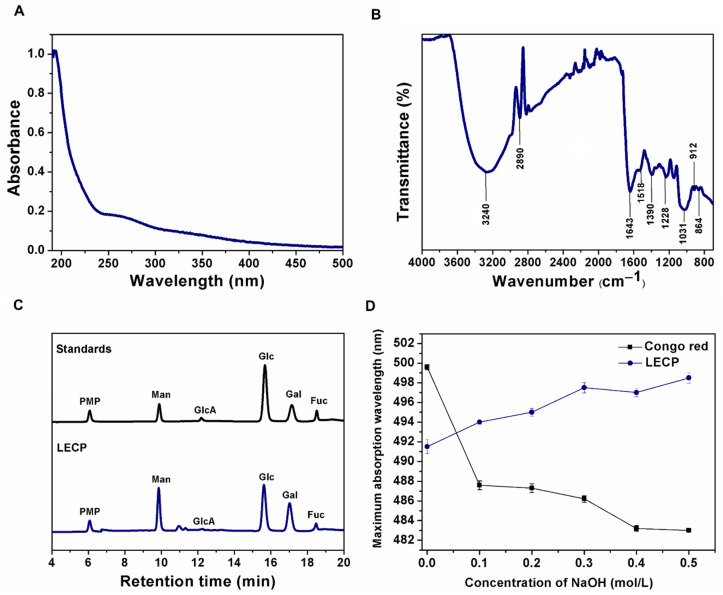
Characterization of LECP: (**A**) UV-Vis spectrum, (**B**) FT-IR spectrum, (**C**) HPLC chromatogram profile of monosaccharide composition, and (**D**) Changes in maximum absorption wavelength of Congo red (control) and Congo red with the LECP at different concentrations of NaOH. The abbreviations can be found in the Section 2.1.

**Figure 2 antioxidants-11-01770-f002:**
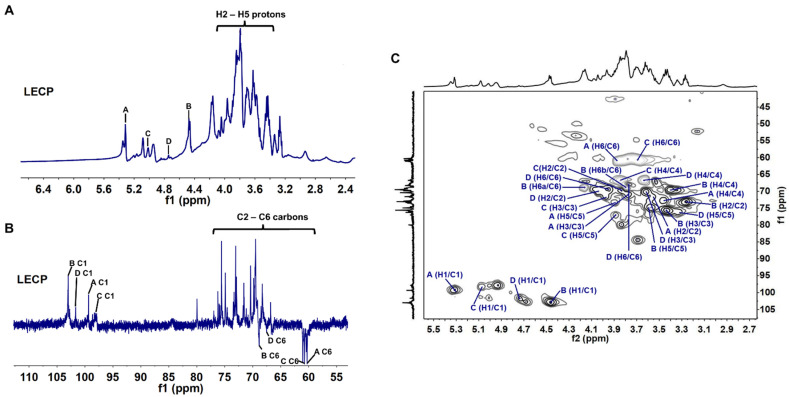
(**A**) ^1^H, (**B**) DEPT-135 and (**C**) HSQC spectrum of LECP (A, B, C and D refer to the residues in the structure of LECP. A: α-D-Glc; B: →6)-β-D-Glcp-(1→; C: →6)-α-D-Galp-(1→; D: β-D-Manp-(1→).

**Figure 3 antioxidants-11-01770-f003:**
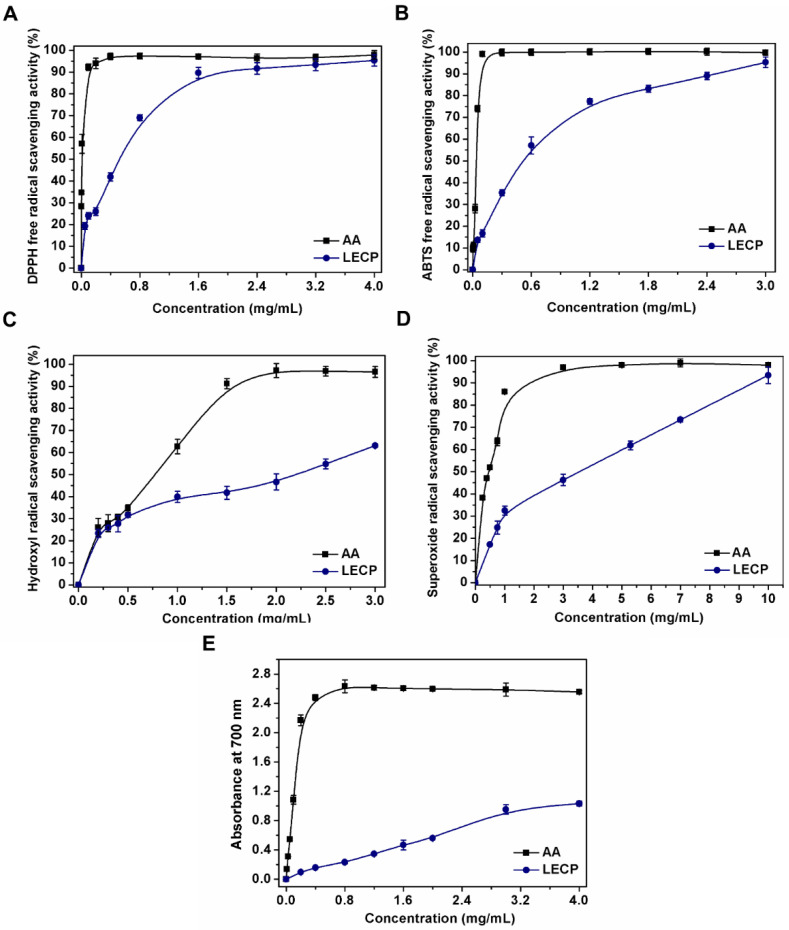
Antiradical activity of LECP and AA: DPPH radical scavenging activity (**A**), ABTS radical scavenging activity (**B**), hydroxyl radical scavenging activity (**C**), superoxide radical scavenging activity (**D**) and reducing power (**E**).

**Table 1 antioxidants-11-01770-t001:** Information about the chemical characterization of LECP.

Parameter, %, *w*/*w*	Reference Substance	Regression Equation, Correlation Coefficient
Neutral sugar	Glucose	y = 0.010x − 0.006, R^2^ = 0.999
Uronic sugar	Glucuronic acid	y = 0.008x + 0.027, R^2^ = 0.996
Sulfated sugar	Chondroitin sulfate	y = 0.006x + 0.011, R^2^ = 0.997
Protein	Bovine serum albumin	y = 0.002x + 0.001, R^2^ = 0.998
Total phenols	Gallic acid	y = 0.088x − 0.012, R^2^ = 0.996

**Table 2 antioxidants-11-01770-t002:** ^13^C and ^1^H NMR chemical shift data for sugar residues in *L. edodes* extracts.

Glycosyl Residues		Chemical Shifts (ppm)					
		H1/C1	H2/C2	H3/C3	H4/C4	H5/C5	H6a, H6b/C6
A	α-d-Glc	5.31	3.56	3.78	3.48	3.89	3.78
		99.29	71.61	70.83	72.78	73.17	60.3
B	→6)-β-d-Glcp-(1→	4.46	3.26	3.45	3.39	3.59	4.16, 3.8
		102.8	73.17	75.51	69.66	74.76	68.88
C	→6)-α-d-Galp-(1→	5.06	3.83	3.97	3.83	3.89	3.69
		98.12	69.66	71.61	66.54	77.07	60.69
D	β-d-Manp-(1→	4.74	4.05	3.61	3.64	3.34	3.78
		101.63	70.05	70.05	66.54	76.29	67.71

**Table 3 antioxidants-11-01770-t003:** Antiradical activity of *L. edodes* extract and AA.

Crude Extract	EC_50_ (mg/mL)				
	DPPH	ABTS	Hydroxyl Radical	Superoxide Radical	Reducing Power
LECP	0.51 ± 0.05	0.52 ± 0.02	2.19 ± 0.18	3.59 ± 0.06	1.73 ± 0.02
AA	0.010 ± 0.001	0.040 ± 0.001	0.79 ± 0.02	0.46 ± 0.03	0.042 ± 0.002

**Table 4 antioxidants-11-01770-t004:** Results of antioxidant activity of LECP and AA.

	CDH (mmol/Kg MeLo)	MDA (mmol/Kg MeLo)
LECP	14.36 ± 2.41	0.08 ± 0.02
BHT	30.68 ± 3.43	0.06 ± 0.01
MeLo	163.72 ± 10.16	10.42 ± 0.95

CDH: conjugated diene hydroperoxide; MDA: malondialdehyde; BHT: butylated hydroxytoluene.

## Data Availability

The data presented in this study are available in the article.

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
