# Peer review of "Lentinula edodes, a Novel Source of Polysaccharides with Antioxidant Power"

_antioxidants, 2022, doi:10.3390/antiox11091770_

Round 1

Reviewer 1 Report

Please use Melo or MeLo how do you prefer but just one see Pg. 2 row 75 and Pg 5 row 214.

Pg. 2 row 88 “The residue was extracted twice with distilled water (1:20 w/v)” Please 1 part, in weight is refer to? Please add the compound.

In the paragraph 2.3.1. Sugar, protein and phenols composition add Table 1 in the text

There are two Table 4, the table pg 12 could be Table 5   

Author Response

Please use Melo or MeLo how do you prefer but just one see Pg. 2 row 75 and Pg 5 row 214.

We decided to use MeLo, and the manuscript was adjusted on pgs. 2,6, 11, and 13.

Pg. 2 row 88 “The residue was extracted twice with distilled water (1:20 w/v)” Please 1 part, in weight is refer to? Please add the compound.

We included this information in the manuscript on pg. 2 rows 89-90.

In the paragraph 2.3.1. Sugar, protein and phenols composition add Table 1 in the text

In the manuscript on pg. 3 rows 133-134, we included Table 1 in the text according to suggested by the reviewer.

There are two Table 4, the table pg 12 could be Table 5 

Thank you very much for the recommendation. The change was made in the manuscript on pg. 13.

Reviewer 2 Report

The manuscript “Lentinula edodes, a novel source of polysaccharides with antioxidant power” by Muñoz-Castiblanco et al. investigated the composition, antioxidant and radical scavenging properties of polysaccharides extracted from Lentinula edodes fritting bodies.

This manuscript presents a piece of visually clear information, sound design experiments, and exciting results. Here I have some comments for the authors:

-         In the abstract, the authors should generally reduce the number of significant figures. Be careful with the use of “,” and “.” for decimal numbers.

-        Section 2.3.1 should include an extended explanation of methods.

-        In section 3.1.1. the table is unnecessary since the authors repeat the same results in the text.

-        Where is the concentration for individual neutral sugars and uronic acids composing LECP? The authors should quantify those components.

-        It is expected that the authors would include more information on the chemical structure of polysaccharides. What polysaccharides are conforming LECP mixtures? B-glucan and chitin? What is the molecular weight of those polysaccharides?

-        When considering antioxidant properties, the authors should also consider using more physiological models to assess radical scavenging properties. For instance, they could evaluate the antioxidant capacity after simulated gastrointestinal digestion and the cellular antioxidant properties in intestinal cells, considering that those polysaccharides would exert their antioxidant properties throughout the gastrointestinal tract.

-        Again, in terms of the antioxidant properties, it would be interesting to compare effects with other polysaccharides that are already used and considered antioxidants instead of using ascorbic acid (or using both) since the latter is a small molecule, which may have completely different mechanisms of action as those polysaccharides.

-        In general, there is little to no discussion. The authors should improve that part of their manuscript, being more critical with their results and better comparing with other polysaccharides, and highlighting the implication drawn from their results and conclusion.

-        The conclusion section should be rewritten. No results should be included there, nor a summary of results. A brief results outline may present conclusions, but this section should give no numbers.

Author Response

The answer is in the attached file

Round 2

Reviewer 2 Report

The authors have adequately answered the reviewer's suggestions. They have not included more experiments, but a more detailed discussion is provided. Then, the manuscript can be now considered for publication.